# An Approach of Social Navigation Based on Proxemics for Crowded Environments of Humans and Robots

**DOI:** 10.3390/mi12020193

**Published:** 2021-02-13

**Authors:** Marcos Daza, Dennis Barrios-Aranibar, José Diaz-Amado, Yudith Cardinale, João Vilasboas

**Affiliations:** 1Electrical and Electronics Engineering Department, School of Electronics and Telecommunications Engineering, Universidad Católica San Pablo, Arequipa 04001, Peru; marcos.daza@ucsp.edu.pe (M.D.); dbarrios@ucsp.edu.pe (D.B.-A.); jose_diaz@ifba.edu.br (J.D.-A.); 2Instituto Federal da Bahia, Vitoria da Conquista 45078-300, Brazil; joaopfvilasboas@gmail.com; 3Universidad Simón Bolívar, Caracas 1086, Venezuela

**Keywords:** mobile robots, social navigation, social momentum, proxemics, proxemic interactions

## Abstract

Nowadays, mobile robots are playing an important role in different areas of science, industry, academia and even in everyday life. In this sense, their abilities and behaviours become increasingly complex. In particular, in indoor environments, such as hospitals, schools, banks and museums, where the robot coincides with people and other robots, its movement and navigation must be programmed and adapted to robot–robot and human–robot interactions. However, existing approaches are focused either on multi-robot navigation (robot–robot interaction) or social navigation with human presence (human–robot interaction), neglecting the integration of both approaches. Proxemic interaction is recently being used in this domain of research, to improve Human–Robot Interaction (HRI). In this context, we propose an autonomous navigation approach for mobile robots in indoor environments, based on the principles of proxemic theory, integrated with classical navigation algorithms, such as ORCA, Social Momentum, and A*. With this novel approach, the mobile robot adapts its behaviour, by analysing the proximity of people to each other, with respect to it, and with respect to other robots to decide and plan its respective navigation, while showing acceptable social behaviours in presence of humans. We describe our proposed approach and show how proxemics and the classical navigation algorithms are combined to provide an effective navigation, while respecting social human distances. To show the suitability of our approach, we simulate several situations of coexistence of robots and humans, demonstrating an effective social navigation.

## 1. Introduction

The continuous evolution of social robotic has fostered the presence of mobile robots in many contexts of people’s daily lives; therefore, leading the evolution of service robots. Nowadays, service robots provide support for specific tasks, such as helping elderly [1,2], giving academic classes to children [3,4], helping as guides [5,6,7], among many other tasks [8,9]. In particular, in such indoor environments (e.g, hospitals, schools, banks and museums), where the robot coincides with people and other robots, its movement and navigation must be programmed and adapted to robot–robot and human–robot interactions.

Autonomous navigation is limited to avoid obstacles, while the robot is reaching the goal. Navigation of service robots should consider factors such as human comfort, naturalness and sociability [10]. To ensure the human comfort, the way the robot navigates must give humans the feeling of security [11], in the sense its trajectory does not impede the natural trajectory of humans. That security is achieved simply with the robot’s evasion of humans; however, the robot trajectory may still be rough, causing humans a feeling of insecurity and discomfort. Naturalness refers to the robot executing movements similar to humans, during a trajectory. Most of the methods that try to match such trajectories, adjust the robot’s speed by smoothing its movements between successive points on the path [12,13]. Finally, the sociability of the robot dictates social behaviours of robots, linked to regional or ethical notions, such as keeping social distances with humans, avoiding to interrupt a conversation among people by passing through them.

This social interaction between humans and robots demands special attention, in environments where both cooperate or work independently, in order to make the behaviour of robots efficient and socially acceptable [14,15,16]. In this sense, social navigation is a crucial aspect to consider for robots being part of habitats and work spaces of humans. Thus, the development of social robots, and their safe and natural incorporation into human environments, is an essential and complicated task, which must inevitably consider the distance [17,18]: misuse of distancing can generate a disruptive attitude of humans towards the robots. The way a robot moves reflects its intelligence and delineates its social acceptance, in terms of the perceived safety, comfort and legibility.

Researchers are therefore seeking to develop new flexible and adaptable interactions, in order to make robot behaviour and, in particular navigation, socially acceptable. In this sense, *proxemic interaction* is becoming an influential approach to implement Human–Robot Interactions (HRI) [19,20,21,22,23]. The original concept of *proxemics* was proposed by Edward T. Hall in 1966, who first presented the proxemic theory [24], which describes how individuals perceive, interpret and use their personal space relative to the distance among themselves [25]. Social relationships are essential in the life of human beings and can be expressed as how people allow contact and interact among each other in a physical space. Thus, people interactions are based on physical distances or face orientation of others. Both factors describe the level of engagement among people to establish communication. This social science theory has inspired researchers to create seamless interactions between users and digital objects in a cyber environment (ubicomp), this is called proxemic interaction. Thus, proxemic interactions describe relationships among people and digital objects, in terms of five physical proxemic dimensions: Distance, Identity, Location, Movement and Orientation (DILMO), and determine proxemic behaviours, i.e., the interactions as response of such digital objects [26,27,28].

As an acceptable social navigation is based on factors, among which the distance from the robot to people is an aspect to take into account, proxemic interaction is an appropriate approach to establish social behaviours of robots, in terms of DILMO dimensions. Therefore, not only distance can be considered, but identity, location, movement and orientation of humans.

In scenarios where mobile robots interact with humans and other robots, their social behaviour must be adapted to robot–robot or human–robot interaction, accordingly. For robot–robot interaction, traditionally navigation algorithms for multi-robot systems are considered [12,29,30,31]. However, there is still a lack of attention on considering the integration of human–robot and robot–robot interactions, and adapting the robot’s behaviour accordingly.

In this context, we propose a novel approach to adapt the navigation of social robots to different scenarios, by integrating proxemic interactions with traditional navigation algorithm. In scenarios where the robot is alone, it performs ORCA algorithm to avoid obstacles.

In the presence of humans, proxemics is combined with A* and Social Momentum, to achieve social behaviours, by taking into account the social restrictions that its actions entail. With this novel approach, the mobile robot adapts its behaviour, by analysing the proximity of people to each other, with respect to it, and with respect to other robots to decide and plan its respective navigation, while showing acceptable social behaviours in presence of humans. We describe our proposed approach and show how proxemics and the classical navigation algorithms are combined to provide an effective navigation. We simulate different scenarios of coexistence of robots and people to validate our proposal, demonstrating the feasibility of an adaptable and suitable social navigation system.

In summary, the main differences of our proposed approach with respect to existing works are (i) the integration of both human–robot and robot–robot interactions in the same approach, (ii) the consideration of proxemic zones for robots in scenarios in which the robots are interacting with humans, (iii) detection of individual and groups of humans to accordingly decide the navigation, and (iv) integration of traditional navigation approaches with proxemics, by considering all DILMO dimensions to accordingly behave during navigation.

The remainder of this work is organised as follows. Section 2 shows some preliminary concepts of proxemic theory, in addition to the social constraints that can occur in an environment surrounded by people. Section 3 details some studies related to autonomous robot navigation, as well as works done with HRI based in proxemics. Section 4 describes our proposal, which focuses on the development of a navigation system for social robots operating in environments populated by both humans and robots. Section 5 shows our navigation system implemented and tested in different situations. Section 6 discusses improvements that can be carried out. Finally, we draw conclusions in Section 7.

## 2. Proxemic Theory and Proxemic Interactions: Preliminaries

Social distances are present in our daily existence. The theory of proxemics is a concept used mainly to describe the human use of space. Edward T. Hall proposed the first definition, who pointed out proxemics as “the interrelated observations and theories of humans use of space as a specialised elaboration of culture” [24]. He presented how people perceive, interpret and use space, especially related to distance among people [25]. The theory of proxemics describes how people from different cultures not only speak diverse languages, but also inhabit different sensory worlds. In this respect, the distance has an indispensable role in proxemics in order to establish a region around the person that serves to maintain proper spacing among individuals.

According to Hall’s theory of proxemics, the interaction zones have been classified into four proxemic zones, as shown in Figure 1:**Intimate zone**, defined by a distance of 0–50 cm (0–1.5 feet). This space is reserved for close relationships among people, and physical contact is possible in this zone. Usually, people can access to this zone if the other person allows it (Figure 2a). An invasion of this zone, without a justification, can be interpreted as an attack, generating discomfort. However, there are exceptions according to environments, such as public transportation and lifts, where the person’s intimate zone can be compromised.**Personal zone**, delimited by distances from 0.5m to 1 m (1.5–4 feet). In this zone, people could have a natural interaction with other people, and it is barely possible to reach contact with their arms in which physical domination is limited (Figure 2b). Beyond it, a person can not freely “get their hands on” someone else.**Social zone**, determined by distances between 1 m and 4 m (4–12 feet). This area can be related to space where people could maintain communication without touching each other (e.g., a meeting table). In this zone, people keep physical distancing among individuals. For example, a business meeting can describe the zone in which people have to speak louder to address others in order to catch their attention (Figure 2c).**Public zone**, with distance greater than 4 m (greater than 12 feet). It describes the distribution of people in urban spaces, such as a concert hall or public meeting, where the people’s attention is focused on a moderator (Figure 2d) or in the street, in parks or museums, where the person is unaware of others. Other people’s identities are unknown among individuals that share the same space. This space sometimes varies depending on the situation. For example, at a concert or on public transportation, people are standing next to each other; in these exceptional cases, social, personal and even intimate zones are invaded. However, it is not considered an invasion as it is aware that the situation does not allow taking the corresponding distance. In other words, sometimes public distance becomes temporarily personal or even intimate space.

Although the proxemic zones of Hall are concentric circles, other studies have proposed different configurations that respond to difference aspects, such as culture, age, gender, personal relationship and context [22]. Figure 3 shows different configuration of proxemic zones [22]. Figure 3a shows the classical four proxemic zones of Hall’s theory, defined as concentric circles. According to the study presented in [32], people are more demanding with respect to their frontal space, considering frontal invasions more uncomfortable. Thus, egg-shaped proxemic areas, as shown in Figure 3b, are more appropriate. In public environments, the personal space refers to the “private sphere” in the Social Force model; thus, the movement carried by pedestrians is influenced by other pedestrians through repulsive forces [33]. Therefore, defining proxemic zones as concentric ellipses (see Figure 3c) seems to be more suitable. In [34], authors performed a study to state that personal space is asymmetrical, as shown in Figure 3d: it is smaller on the dominant side of the pedestrian (right-handed or left-handed). The study demonstrates that when people want to go through an insufficient space, they first evaluate the relationship between the size of the passage and the width of the body.

In crowed environments, people’s interactions are not only limited to person-to-person. Groups of persons also change the way an individual interacts with groups in which he/she is not participating (e.g., to do not pass through the group). Thus, it is also important to define proxemic zones for group of people. Results presented in [35] demonstrate that people fix more space around a group than with the sum of individual personal spaces. The concept of the “O” space, proposed in [36], allows detecting conversations. The “O” space is the area that delimits the main activity established by a group of people (e.g., having a conversation or focused in a common object or situation). Only participants can enter it, they protect it and others tend to respect it, and its geometric characteristics depend on the size of the body, posture, position and orientation of the participants during the activity. Orientation of people can help deciding which groups are conversing and how the space “O” is defined. The “O” space is extended with the “P” space in [36], which surrounds the space “O” to locate participants and their personal belongings. Figure 4 shows the “O” and “P” spaces in white and red circles, respectively.

Besides the individual and group proxemic zones, human activity can define other virtual spaces that others recognise and respect, such as activity and affordance spaces. The activity space is a social space related to the action carried out by a person. The affordance space is a social space related to a potential activity provided by the environment. In both cases, the notion implies a geometric space, but it does not give an explicit definition of the form, as it can take many forms depending on specific (potential) actions. Figure 5a shows an example of an activity space, in which the human is taking a picture; the surrounding people should avoid this space in order to not interrupt the activity. In Figure 5b, the space in front of the picture can potentially be used to read the information; thus, people should avoid to occupy this space.

Researchers have been inspired by this social science theory to create seamless interactions among users and digital objects in ubicomp environments. The concept of proxemic interaction was first proposed by Ballendat et al. in [26] and was designed for implementing applications in ubicomp environments (see Figure 6). Ubicomp has included different services and smart technologies (Internet, operating system, sensors, microprocessors, interfaces, networks, robotics and mobile protocols), which allow people interacting with the environment in a more natural and more personalised way [37]. Therefore, ubicomp provide new opportunities to explore new approaches for Human–Computer Interactions (HCI), in environments where users have many computing devices that can be employed according to the context required by them.

Greenberg et al. [27] identified five dimensions: Distance, Identity, Location, Movement and Orientation (we call them DILMO as an abbreviation), which are associated with people, digital devices and non-digital things in ubicomp environments. Thus, proxemic interaction has been implemented in order to improve HCI in such as ubiquitous environments, by determining proxemic behaviours, i.e., responses of such digital objects according to DILMO dimensions with respect to people or other objects (digital or not) [38]. In the context of HRI, DILMO dimensions can be used to model human–robot, robot–robot, and robot–device interactions.

Proxemic DILMO dimensions can be analysed in a variety of ways according to the measures that can vary by accuracy and the values they return (i.e., discrete or continuous), which in turn depend on the technology used to gather and process them.

**Distance** is a physical measure used as a parameter to determine the proxemic zone of entities (users, devices and robots), based on Hall’s theory. The zone allows the users to interact with the display, device or robot according to different proximities [27,39]. Typically, short distances allow high-level interactions between devices, between a user and a device, between a robot and a person, between robots, etc.

For social navigation in service robots, determining the human–robot distance, human–human distance and robot–robot distance are important. It can be obtained based on several combination of hardware and software capacities of robots, such as kinect sensor technology, which provides body-tracking and object-tracking capabilities; camera ability, to apply computer vision or thermal imaging analysis; and ultrasonic and infrared sensors, to detect entities at certain distances.

**Identity** is a term that mainly describes the individuality or role of a person or a particular object that distinguishes one entity from another one in a space [27,40]. The identity can be used, for example, for controlling spatial interactions between a person’s handheld device and all contiguous appliances in order to generate an effective appliance control interface [38] or to display specific content for a specific person (e.g., for access control, content in TV for children is different to the content for parents).

For service robots, **Identity** can be used to determine, for example, specific people with specific roles or a specific device. Kinect sensors and cameras of robots can be used to find out identities.

**Location** defines the physical context in which the entities reside. The location allows relationships of entities with objects which are categorised as fixed (e.g., room layout, doors and windows) and semi-fixed objects that are changeable such as chairs, desks, lamps [27,38]. It is an important factor because other measures may depend on the contextual location. Location provides the entity’s positions in the space, that are assessed at any time.

Almost all robot technologies used to get Distance are also useful to obtain **Location**, as these two measures can be co-related.

**Movement** is defined as entity’s change of positions over the time [27,38]. The movement includes the directionality allowing interaction between the user and the application. For example, when a user walks towards the screen, the content of the screen is adjusted according to the user’s movement speed or when a human is approaching a robot, it decides an action according to the context.

**Movement** can be detected by kinect sensors, leap motion sensors, ultrasonic and infrared sensors. Furthermore, kinect and thermal cameras provide capabilities of detecting movements with computer vision or thermal imaging applications.

**Orientation** provides the information related to the direction in which entities are facing between each other. It is only possible if an entity has a “front face” and the entity can be detected in the visual field of another entity. Orientation can be continuous (e.g., the pitch/roll/ yaw angle of one object relative to another) or discrete (e.g., facing toward or away from the other object) [27].

**Orientation** is a key dimension in HRI and can be determined with kinect depth and thermal cameras, marker-based motion sensors, combined with computer vision, face recognition, and machine learning techniques. According to the orientation of people, a robot can detect group of people, activity and affordance spaces, and people talking to it.

## 3. Related Work

The autonomous navigation of mobile robots is a great challenge in the academic area, especially in dynamic environments, with the presence of humans and objects that move or change positions at any time. In order to solve this navigation problem, there are works that try to predict the movement of obstacles, imitate movements of animals, predict routes of people or objects and map the place of performance [41,42,43]. In order for the robot to predict the routes and movements of humans or objects involved in the environment, it is necessary to incorporate sensors, which help in this process, such as scanners [41], tracking sensors, motion capture cameras, etc. Using the sensor data, robots trace the routes that they can follow, while mapping the environment. This planning can be divided into two forms: global or local.

When performing a global planning, the robot usually has a static map, which implies that it already knows the environment while moving. Local planning work with dynamic situations and generally does not know the initial map. These two types of planning commonly work together in a navigation process. Some works are detailed below. In [44], a dynamic local navigation plan is proposed, through the modulation of artificial emotions present in two robots. In [45], a navigation system is described for static and dynamic environments, imitating the behaviour of ants when they move from one point to another. A robot signals the environment with artificial pheromones all the way, to use as a reference for it and other robots. Among the navigation algorithms, it is common to find those that plan the construction of virtual maps, for their correct navigation, such as Simultaneous Localisation and Mapping (SLAM) [41] and the Map Server Agent (MSA) [46], that propose to keep the graphic model of the environment, together with the positions of each object, while searching for the shortest path to arrive at the goal.

In scenarios in which the mapping and the perception of moving objects is performed, the next step is to avoid collisions and respect the human space. In [47], a mathematical algorithm is proposed, that with sensors assistance, produces fast and realistic simulations with, at first, obstacle speed information, generating a field around it to prevent a path with collision. This algorithm is the Optimal Reciprocal Collision Avoidance (ORCA). However, there is still a lack of discussion to improve HRI, as with ORCA, robots perform very well in inhospitable works. Nowadays, robots are being developed to act in services with human contact [48], as the autism therapeutic treatment [49] or museum guides [50]. Therefore, recent studies are focused on improving such HRI, by considering proxemic theory to respect human’s proxemic zones [31,51].

The study presented in [19] conducts experiments to observe interactions among people and robots and extracts some notes, such as the higher mutual visual contact that was obtained, the higher distance between the person and robot; the more likeable the robot, the shorter this distance; men keep higher distance than women, and even more so when the robot make visual contact; also, the human–robot distance was shorter when the human was facing the robot’s back. Another experiment is described in [52], with three different algorithms to model interactions between 105 people and a robot. The best result was obtained with the Social Momentum algorithm, compared with ORCA and a tele-operation approach.

Social Momentum is a cost-based planner that detects an agent’s intentions about collision evasion protocols. This cost function is the weighted sum of the magnitudes of the angular moments, both of the planning agent (robot), and of the other entities in its environment (humans). This type of indirect communication of avoidance strategy results in easy-to-interpret movements, allowing the robot to avoid entanglements in its path. In an area overrun by many agents, whether people or robots, the planning agent interacts with each other, in the sense that each action that the robot takes transmits signs of intentions or preferences about evasion strategies (move right or left). Social momentum makes the robot read the movement preferences and associate them with its own. Social Momentum allows the robot to act pertinently and simplify everyone’s decision-making. In conclusion, Social Momentum is a frequent replanning algorithm in which in each planning cycle it determines an action [53]. If the robot identifies a future fault and decreases its velocity, its inactive time will be shorter, but if the robot does not cushion its path, it will be parked for a considerable time, which will cause a human discomfort in robot’s presence. The ideal is that the robot does not invade the intimate human space and keeps a constant and soft velocity, but not too slow to create a discomfort or get its task late. Social interaction spaces are widely modelled by using Gaussian functions [54].

As well as the global and local planning may be merged to more complete planning elaboration, it is also possible to combine Social Momentum and ORCA. In [12], the velocity is an important aspect in Social Momentum. Observing the experiences, Social Momentum generates a constant and cushioned speed to the human route. Some works have analysed what is the limit of the robot speed that does not take the human comfort out, and also does not take the task efficiency out. In [22], the authors explain some models of human behaviour that could be implemented in a social navigation and some models based on proxemics. Authors conclude that in several cases it is not enough to analyse the human trajectory. In [55], some particular cases are presented in which the robot is forced to interact with humans through a chatbot API. This work analyses and places phrases that the robot can use to maintain verbal communication during its navigation. In [56], the use of social robots focused on the workplace is addressed; the surveys investigate the effects of the presence of robots in an area normally inhabited by humans. In [57], a planner is presented that is capable of predicting the trajectories followed by humans and in turn planning the future trajectory of the robot. Considering that the position of the objects is very significant, in [58] an analysis is given on how a robot should position itself when facing a certain person, for this the research is based on proxemic principles. In [59], the theory of equilibrium is used, which analyses the impact that coexistence between humans and robots has generated. In [60], an analysis is done on the proxemics that drones must respect for people. Researchers consider new distances between the human–drone interaction. In [61], authors propose a trajectory planning system that takes into account the time of day and the possibility or not of using some spaces in specific periods of time. The propose a method that restricts or penalises the route planned by the robot, due to time-dependent variables. The authors also present an example of a case in which a patient undergoes a physiotherapy section, by used a feature table for this purpose. In this case, considering the scheduling of the referred physiotherapy section, the robot must plan a route over a distance limit greater than usual; thus, the usability and proxemics limits are respected. This social information is added to a graph and is used later for trajectory planning.

Another approach to model the social space is proposed in [62]. Authors present a new definition of social space, named as Dynamic Social Force (DSF). This new social space definition is based on a fuzzy inference system and the parameters of these functions are adjusted by using reinforcement learning. They use reinforcement learning to determinate parameters for the Gaussian function. Despite that, it is not investigated the proposed method to generalize group of people.

Table 1 summarises a comparison of the revised studies, some of which have considered proxemics to offer a social navigation. We compare them in terms of DILMO dimensions, if they are able to detect individual or group of people and the considered proxemic zones. Most studies consider distance and orientation, as well as individual personal zones. Few works consider other DILMO dimensions, such as location and movement, and proxemic zones for groups of people. Our proposed social navigation system is aimed at considering all DILMO dimensions and proxemic zones for individuals and groups of people.

The main differences of our proposed approach with respect to existing works are (i) the integration of both human–robot and robot–robot interactions in the same approach, (ii) the consideration of proxemic zones for robots in scenarios in which the robots are interacting with humans, (iii) detection of individual and groups of humans to accordingly decide the navigation and (iv) integration of traditional navigation approaches with proxemics, by considering all DILMO dimensions to accordingly behave during navigation.

## 4. Navigation with Proxemics: Our Proposal

Social navigation for autonomous robots demands practitioners and scientists to rethink the traditional strategies, to ensure seamless integration of robots into social environments. In this context, proxemic zones plays an important role to model social navigation.

In social environments, robots can coincide with a person, with groups of people or with groups of people with other robots. Thus, considering only individual proxemic zones is not enough to determine the human use space. The treatment of proxemic zones should allow defining dynamically the human use space as well as distinguishing proxemic zones for individuals and for groups. The spatial patterns adopted by people in conversations act as social cues to inform robots about their activity, the orientation of people can help decide which groups are conversing and where the space “O” would be located. Social robots could benefit from that knowledge to identify social interactions in indoor environments.

Moreover, activity and affordance spaces (e.g., looking at pieces of art or informational posters in museums) might also be considered by robots. Consequently, several recent works focus on developing a more adequate navigation system in terms of human comfort.

In addition, in social spaces, it is possible to have only robots working alone without human interaction. In this scenario, it is not necessary or advisable to use proxemics for robots because they are only machines and can be considered mobile obstacles; thus, traditional techniques of collision avoidance with mobile obstacles or between robots must be used. This means that robots should be able to adapt their behaviour according to the specific situations.

The proposal in this research is focused on the development of a navigation system for social robots acting in environments populated by both humans and robots. The proposed approach considers that the robot must avoid collisions in a social environment and incorporates restrictions based on proxemics, in particular for robot–human interaction for the greater comfort of people.

The traditional flow of a navigation system carries out, in a sequential flow, the following tasks (see Figure 7a): Perception, Localisation, Path Planning, Path Execution, Path Adjustment and Low-Level Control. It is related to the control architecture implemented on it. The flow starts with the perception of the world by the robot, including the goal position and obstacles (static and mobiles); next, the robot must perceive its own position in order to plan the path it needs to follow. Finally, the robot must execute the path by transforming it in low level actions for executing it in its actuators.

The proposal in this paper is based on modifying the flow between the execution of the path and its associated low-level actions. In Figure 7b, the three modifications to the traditional flow are shown. The objective is to improve the path planned by the robot, obtained from its path planning algorithm, by adding social or robot avoidance restrictions (depending on the situation). Once the robot has defined its path to reach the goal, it will begin to navigate the environment. As the robot is in a social environment, the correct execution of the defined path may be conditioned by the presence or absence of human beings.

During the execution, the robot can find itself in several specific situations, which imply presence of other robots without humans, human presence, and human and robots presence. In the presence of human beings, their behaviour must be socially acceptable. That is, if the robot does not have some social restrictions, it can generate discomfort among people, causing a rejection of its insertion into the environment. A socially acceptable behaviour includes, for example, that the robot must not pass between two people who are talking, must avoid approaching someone abruptly, must not get too close to someone (invading the intimate proxemic zone of the individual), among other situations. Likewise, in an environment where there is interaction between humans and robots, the breakmaker cannot interrupt that interaction, much less pass too close to the group. On the contrary, when there are no human beings in the robot’s environment, its navigation will not be conditioned to social behaviour, but it must consider cases of collisions with objects or other robots in the environment. From there, two possibilities of the proposal are derived: the use of proxemics for situations that include people presence and the use of only obstacle avoidance algorithms when there is no presence of human. Figure 7b shows the flow of the proposed approach.

In this proposal, proxemic zones are modelled with Gaussian functions. The next sections describe in detail how human–robot and robot–robot interactions are implemented in our proposal. To illustrate its functionalities, we present examples in the context of service robots in museums.

### 4.1. Navigation with Absence of Humans and Presence of Other Robots

Navigation with the sole and exclusive presence of other robots is widely studied today. There are many techniques and algorithms that solve this problem. Techniques range from speed control, location using particle filters, SLAM-based algorithms, graph-based techniques, nature-based algorithms and so on.

This aspect of the proposal considers the scenarios in which the robot shares the navigation environment with other robots, but is not surrounded by humans (see Figure 8 as an example). If the robot perceives that there are only robots co-working with it, the others become dynamic objects. In this case, the robot does not need to follow or use social rules, it must exclusively use dynamic obstacle avoidance rules. Therefore, for this part, it is necessary to have an algorithm or technique for adjusting the planned path; giving the robot the capability of avoiding collisions with other agents all the time.

Our approach includes the use of the Optimal Reciprocal Collision Avoidance (ORCA) algorithm [63]. The ORCA algorithm provides a sufficient condition for several robots to avoid colliding with each other, that is, it guarantees collision-free navigation in real-time. It is a local collision avoidance approach based on the notion of obstacle speed (*VO*). This algorithm assumes various robot conditions. It assumes that the robot is holonomic, that is, it can move in several directions. Furthermore, it assumes that each robot has perfect detection and is able to infer the exact shape, position, and speed of other robots in its environment.

The task of each robot is to select (independently) a new speed (*vnew*) to guarantee that there will be no collision, at least for a short period of time (τ). By varying the speed, ORCA ensures that there will be no collision between robots. Therefore, the ORCA algorithm is based on the persistent effort to keep the robot collision-free, requiring only to know the radius, position and speed of each robot.

This algorithm is chosen to be included in the planning improvement proposal due to its good results when navigating with other robots. A research carried out with this algorithm shows that ORCA generates a smooth movement, with less acceleration, with less irregularity of the trajectory and greater energy savings in contrast to other methods [52].

### 4.2. Navigation with Only Humans Presence

As explained in Section 3, social robotic navigation in human-populated environments is an area that starts to evolve considering the Hall’s theory of proxemics [24] and five proxemic dimensions [27]: Distance, Identity, Location, Movement and Orientation (DILMO).

When the robot perceives the presence of human beings, it adopts social restrictions in its navigation, thus respecting the rules of proxemics, giving main interest on avoiding to invade intimate or personal spaces of humans when navigating the path. Then, the robot adopts social characteristics of the human when it is inserted under a group of human beings, that is, the robot will navigate on its planned route, avoiding colliding with others, but adding the consideration and respect for the proxemic zones of people, especially avoiding invading intimate ones.

Given all this, it is proposed to select, integrate and adapt traditional planning algorithms, in order to get appropriate human–robot and robot–robot interactions. Through the research carried out, we select and integrate A* and Social Momentum algorithms, which are suitable for these cases. Social Momentum is a cost function-based scheduler that detects agent intent on collision avoidance protocols. This cost function is the weighted sum of the magnitudes of the angular moments, both of the planning agent (robot) and of the other entities in its environment (humans). This indirect communication of the robots’ avoidance strategies results in behaviours that are easy to interpret and therefore allows the robot to cooperate implicitly in avoiding entanglements in the trajectory.

Within a space full of multiple agents, the robot interacts with one and the other, in the sense that each action taken transmits signals of intentions or preferences regarding evasion strategies (go right or go left). Social Momentum allows the robot to read those movement preferences and associate them with its own and act competently to simplify everyone’s decision-making. Thus, the algorithm is based on frequent re-schedulers, in which, on each planning cycle it selects a certain action.

For instance, when a robot is in front of a human being (see Figure 9), it is necessary that the robot not only avoid the human, but also respect her/his proxemic zones, by not passing too close. The robot will take into account the intimate proxemic distance to be respected. In addition to this, the DILMO dimensions are always present, such as Location to know if they are in a closed corridor or are in an open environment, Movement to detect if the person is going towards it or away from it and Orientation to detect if the person is in front of the robot, if it is on its back or if the person is looking at an object. All these parameters are important to have a correct social navigation. In this case, when the robot is face to face with a human, it is proposed to use the Social Momentum algorithm so that the robot avoids the person by surrounding it and taking the opposite direction from which the person is heading.

Another possible situation to occur is when the robot runs into a group made up of humans. In this case, the robot cannot pass between them, as it would interrupt their interaction, nor could it pass very close to the group or obstruct someone who wants to join that group (see Figure 10). For this situation, the robot will also have to take into account the DILMO dimensions: The Distance between people to determine the groups and not interfere in their personal or intimate area, the Movement to know if the group moves or stays in one place, the Orientation to detect if the group is looking at a painting, piece of art or something of interest. In this particular case, it is possible to model a group by using very close individual Gaussians (for representing the proxemic zones), almost forming a single one that involves the entire group. This new composition will have to be kept in mind by the robot when navigating to the goal. It will have to consider not only the personal Distance of a human, but of the other people that make up the group.

Another example is the situation shown in Figure 11, where a person is presenting or leading a group of people. This person could be one of the tour guides or someone in particular showing an object (in a touristic place). In this case, the Orientation and Distance are important to take into account, as the robot must not pass through them or pass very close to them as this would interrupt the interaction of the group of people (if it occurs, the situation will become very uncomfortable to people).

The integration of the Social Momentum and A* algorithms are considered in our approach due to its excellent results when tested. This algorithm has shown that, when navigating with humans, it is the best at generating adequate paths, not very complex paths, paths in which the robot reaches its goal faster and paths where more energy is saved. Complementing this, note that when this algorithm was tested in [52], people indicated that they felt much more comfortable with the behaviour of the robots derived from Social Momentum than with another algorithm, because allows to generate smooth movements, facilitating the interpretation of future movements of the robot to avoid colliding with it.

### 4.3. Navigation with Humans and Other Robots Presence

Another situation considered in the proposal is when the social robot shares its space it navigates with people and also other robots (see Figure 12). These secondary robots also have the ability to interact with people, thus forming groups of people and robots. In these cases, it is important to consider areas of proximity to them (very similar to human proxemic zones), which the planning agent will have to respect; because, as the robot interacts with humans it shares their social needs. For this case, each robot will be assigned an area equal to the intimate proxemic zone, only and exclusively when the secondary robot interacts with humans.

To determine if there is human–robot interaction, it must be verified if the secondary robot invades the personal proxemic zone of the humans, if there exist intersection between intimate areas of the robot and the humans, it is conclude that there is an interaction between them. This consideration of assigning the secondary robot a proxemic area is to give comfort to the human, as when the human interacts with the secondary robot it considers it as if it were another human (psychologically speaking). Therefore, when the planning agent has to pass through this group of humans and robots, it should not go too close or in the middle of them.

In this case, as the navigation would be very similar to when the robot encounters a group composed only of humans, the Social Momentum algorithm is used to be in charge of modifying the route traced initially by the A* algorithms in the path planner. This gives the robot a navigation with consideration of social restriction for both the human and the robots that are interacting with humans.

In the following section we show the implementation and results of our proposed social navigation system.

## 5. Social Navigation System: Implementation and Results

We have implemented and tested our proposed social navigation system in a simulator based in Matlab. Different simulation environments have been developed in order to test the proposal. Obstacles were generated in different positions and orientations, as well as people and robots that make part of the environment to each simulation process. Algorithm 1 shows the main structure of the whole process that robots might follow during their navigation. At the beginning, robots trace a possible path from their location to the goal, by using the A* algorithm (line 1), which ensures an optimal solution. This path is followed or modified according to the situations detected during the navigation (line 2). At every moment, robots perform a detection algorithm able to recognise mobile objects, i.e., robots and humans (line 3). The robot’s identity is sent/received, thus they can distinguish robots from humans (line 4). When all detected mobile objects have an identity, robots infer that they are in the situation with only robots presence; therefore, collisions are avoided based in the ORCA algorithm, while reaching the planned path (lines 5 to 11). If one or more persons are detected (e.g., mobile objects without IDs or by face recognition), then the robots’ navigation should follow social restrictions (line 12). To do so, robots are able to detect individuals and groups of people based on their Movement and face Orientation (line 13). When a person is detected, his/her proxemic zones are built, based in the Distance between the robot and the person with a Gaussian; thus, the *Social Momentum* algorithm is executed to avoid the intimate zone of the individual (lines 14 to 16). As the *Social Momentum* algorithm evaluates on each step of the robot if the intimate zone intersects with its way, then the intimate zones of people will never be invaded by robots. A similar procedure is done when a group of people is detected. The proxemic zones of all participants in the group are built with Gaussians, including robots participating in the group that are treated as persons; thus, the “O” space is determined (lines 17 to 19).
**Algorithm 1** Social navigation process1. Run A* algorithm2. **While** robot not in goal position **do**3.     Mobile objects detection (robots and humans)4.     Send/Receive robots’ IDs (robots’ Identity is transmitted/recognised)5.     **if** Only robots are detected (all mobile objects have an Identity) **then**6.         Check Collision Cone7.         **if** there is possible collision **then**8.             Run ORCA setting a partial goal in a point in the planned path calculated by A*9.         **else**10.             Go goal executing planned path11.         **end if**12.     **else if** Humans are detected **then**13.         Detect individuals or groups (based on Movement and face Orientation of people)14.         **if** The nearest mobile object is a person **then**15.             Build a Gaussian for that person (based on Distance)16.             Run Social Momentum setting a partial goal in a point in the planned path calculated                  by A*17.         **else if** A group is detected **then**18.                 Build a Gaussian for all participants in the group, even if there is a robot participating                    in the group and treat it as a person (build the ‘O’ space)19.               Run Social Momentum setting a partial goal in a point in the planned path calculated                    by A*20.         **else**21.               Go goal executing planned path22.         **end if**23.     **else**24.     Go goal executing planned path25.     **end if**26.   **end while**27. **end**

When collisions, robots or people are detected in the planned path calculated by A*, robots should recalculate the path by setting a point in the A* path as a partial goal; therefore, the A* path will be retaken as conditions and social restrictions allow it (line 8, line 16 and line 19). If nothing gets in the planned path, robots continue their navigation as usual (line 10, line 21 and line 24).

### 5.1. Implementation of Proxemic Zones

To represent proxemic areas and distances to each person or robot, when necessary, we use level curves generated by a Gaussian function. They delimit the area where the robot can generate a path planning. Figure 13 shows the projection of a Gaussian function in 3D and Figure 14 illustrates the respective proxemic zones represented by the Gaussian.

### 5.2. Navigation in an Environment Simulation with Robots

Considering that the classifier agent detects that the robot is inserted in an environment with other robots, without the human presence, the ORCA algorithm defines its trajectory to avoid obstacles. In this situation, it is not necessary to use proxemic zones, because only robots exist in the environment (lines 5 to 11 in Algorithm 1). In Figure 15, we show four robots moving to a point in common, without colliding and generating a correct trajectory using the ORCA algorithm.

### 5.3. Navigation in an Environment Simulation with a Person

Figure 16 shows a situation in which a person is moving to the front of the robot. Figure 17 shows the corresponding behaviour of the robot in such as situation. From the classical navigation flow, the A* algorithm (line in blue in Figure 17) generates an optimal path; however, this path does not respect the proxemic zone of the human. Therefore, the Social Momentum algorithm modifies the result of the A* algorithm taking into consideration the proxemic zones of the human (line in red in Figure 17) to generate a safe trajectory for humans. However, the A* path is retaken as soon as the robot does not coincide with the intimate zone of the human (lines 14 to 16 in Algorithm 1).

### 5.4. Navigation in an Environment with Groups of People

Figure 18 shows a situation where a group of people is detected by the robot. In this situation, the intersection of Gaussians representing proxemic zones of each person indicates the possible existence of a group of people. Thus, people orientation helps to determine if people are talking and how the space “O” is defined. The Social Momentum algorithm is used to define the path, respecting the group’s space (lines 17 to 19 in Algorithm 1).

### 5.5. Navigation in an Environment with Groups of People and Robots

Figure 19, Figure 20 and Figure 21 show the behaviour of the proposed navigation algorithm, considering proxemic restrictions in an environment with humans and robots. In Figure 19, a group of two people and a robot is detected by the planner robot. In this situation, a proxemic zone is associated to the robot in the group, which together with the proxemic zones of people form a space “O”. Taking into account the proxemic restrictions, a trajectory is defined by the Social Momentum algorithm. In Figure 20 and Figure 21, we illustrate different scenarios with groups of people and robots, which validate the proposal presented in the generation of paths and respect the proxemic restrictions (lines 18 to 20 in Algorithm 1).

### 5.6. Empirical Evaluation of the Proposed Social Navigation Approach

To evaluate the performance of the proposal, we use two metrics taken from the work in [64]. The first metric is based on the ratio of lengths of two paths ($L_{ratio}$). With this measurement, it is possible to take into account the quality of one path over another, considering the shortest path as ideal. In this work, the path proposed by the path planner A* is the shortest path, whose ratio with the path modified by our algorithm indicates how close are they. Equation (Equation 1) shows $L_{ratio}$, the relationship between the path length of A* ($D_{A*}$) and the path length of our social navigation algorithm ($D_{SN}$).
(1)$$L_{ratio}=\frac{D_{A*}}{D_{SN}}$$

The second metric that we use is based on the smoothness of the movement that the robot has (Smooth). The robot has a smooth trajectory when the mean of all the angles $\theta_{i}$ is closer to zero. This measurement is based on the Mean Square Error (MSE) and is expressed as shown in Equation (Equation 2), where $\theta_{i}$ is the angle obtained when the modified route exits and re-enters the original route provided by A* and *k* denotes the number of angles that will be measured on each route. We use this Smooth metric to evaluate the movements of robots when they leave and retake the A* path. While these movements are smoother, robot navigation is more socially acceptable.
(2)$$Smooth=\frac{1}{k-1}\sum_{i=1}^{k}{\left(\theta_{i}\right)}^{2}$$

Table 2 shows the results for the scenarios with humans previously presented. $L_{ratio}$ values indicate that the modified path is at least 80% similar to the A* path; the average for these cases is 84%; thus, the *Social Momentum* algorithm assures never to invade the intimate zones of people ((line 16 and line 19 Algorithm 1)), while trying to do not deviate too much from the shortest path proposed by the path planner A*. Although Smooth values are more varied than $L_{ratio}$, the average shows that the smoothness of movements that robots perform when avoiding invading the proxemic zones are quite smooth. The best one is obtained in Path 1 (with one person as shown in Figure 17), with 0.08 rad ($4.6^{\circ }$). The highest obtained Smooth value is in Path 3, corresponding to Figure 19, with 0.77 rad (45${}^{\circ }$), this may be due to the position that each person occupies; however, 45${}^{\circ }$ does not represent a rough move. On average of all the tests carried out, we have a smoothness of 0.43 rad, which is equivalent to 25${}^{\circ }$.

## 6. Discussion

The first version and simulation of our proposal demonstrate the feasibility and suitability of a robot navigation system able to be adapted to different situations of humans and robots presence and provide social behaviour to robots. This experience also gives the opportunity of extracting its current limitations and some lessons learned.

### 6.1. Social Rules to Robots

Robot navigation in human-populated environments is a subject of great interest among the international scientific community in the area of robotics. In order to be accepted in these scenarios, it is important for robots to navigate respecting social rules. Avoid getting too close to a person, not interrupting conversations or asking for permission or collaboration when it is required by social conventions are some of the behaviours that robots must exhibit. This paper presents a social navigation system that integrates different software agents within a cognitive architecture for robots and describes, as the main contribution, the corpus that allows to establish behaviours of robots in presence of humans in real situations to improve the human-aware navigation system.

The corpus has been experimentally evaluated by the simulation of different situations in a museum, where robots need to plan interactions with people and other robots. Results are analysed qualitatively, according to the behaviour expected by the robot in the interaction performed. Results show how the corpus presented in this paper improves the robot navigation, making it more socially accepted.

In the current version of the proposed social navigation system, proxemic zones are modelled with the same symetric Gaussian function for people and robots, representing them as concentric circles. However, the parametrisation of proxemic spaces, represented by asymmetric Gaussian, can be defined by the social conditions of the environment, for example, according to culture, gender and customs. Allowing flexibility and improvement of the autonomous navigation system according to social restrictions. Thus, our approach can be improved and extended with the ability of robots to recognize different characteristics of people and accordingly define proxemic zones.

### 6.2. Detection of Groups of People

In the context of social navigation in environments populated by humans and robots, it is also relevant to consider proxemic zones for group of persons. The spatial patterns adopted by people in conversations act as social cues to inform robots about their activity. The current proposal considers Distance, to define proxemic zones; Orientation of people, to decide which groups are conversing and where the space “O” is located; and Identity, to identify robots from people. Considering all other DILMO dimensions, this proposal can be improved in this aspect. Location and Movement can also be perceived by the sensorial capabilities of social robots to detect groups being formed, to define the “P” space, etc.; therefore, to have benefit from that knowledge to identify social interactions in indoor environments.

### 6.3. Activity and Affordance Spaces

In general, an affordance space can be crossed without causing any disturbance, unlike the space of activity, but blocking a space of affordance might not be socially accepted. With the current proposal, activity spaces can be detected based on the proxemic zones and orientation of people. For the recognition of affordance spaces, the perceived geometric characteristics of the environment must be linked with the semantic information of the objects in the environment to achieve a semantic navigation of the robot. However, the task is complicated as the perception of the environment made by the sensors is objective, while the human abstraction of space is very subjective. Then, it is a matter of inferring according to the context and previous knowledge about the environment, which spots of the empty space are restricted to the robot’s navigation. In any of the cases, it is necessary to take into account the semantics of the space when planning socially acceptable navigation solutions.

### 6.4. Inclusion of Other Features in Robots Navigation

In environments populated by humans and robots in the context of social navigation, in fact, we could not take into account only the proxemic zones, as, in the human and robot interaction, there are other features that need to be analysed. In this context, several restrictions related to common spaces, limitations, customs, habits, age, culture, person emotion and the emotions of a group of people will make the navigation system safer and efficient. In this sense, the proposal can be improved by considering all DILMO dimensions, combined with perception and recognition capabilities of robots based on traditional Machine Learning techniques and other planning strategies.

## 7. Conclusions

The need of interaction between machines and humans is becoming more common in people’s daily lives. The effort to improve these relationships through the interpretation of social behaviours are more and more frequent among researchers in area of social robotics. The theory of proxemics is a concept used mainly to describe the human use of space. Thus, carefully designed proxemic behaviours in robots might foster closer human–robot relationships and enable widespread acceptance of robots, contributing to their seamless integration into society. However, when the robot is sharing the same environment only with other robots, it is not necessary to consider social restrictions. Thus, an effective social navigation should adapt to these situations.

In this context, we propose an adaptable and efficient social navigation approach based on the ORCA algorithm to prevents collisions in real-time, in situations with only robots and on the Social Momentum algorithm combined with A* to detect the proxemic zones of humans and robots. Proxemic zones for a robot are defined only if it located inside a human’s personal zone. We use symmetric Gaussians to represent distances and proxemic zones, which can be parametrised depending on personal or cultural characteristics.

In this paper, we show the implementation of our proposed robotic navigation system and illustrate its functionality, by simulating several situations of multi-robot navigation (robot–robot interaction) and social navigation with human presence (human–robot interaction).

For future work, we plan to improve our proposal as we explain in Section 6, for example, by using asymmetric Gaussian function, which will allow modelling the proxemic distances according to the different social characteristics and by considering all DILMO dimensions. We are also looking forward to implement it in a real robot system.

## Figures and Tables

**Figure 1 micromachines-12-00193-f001:**
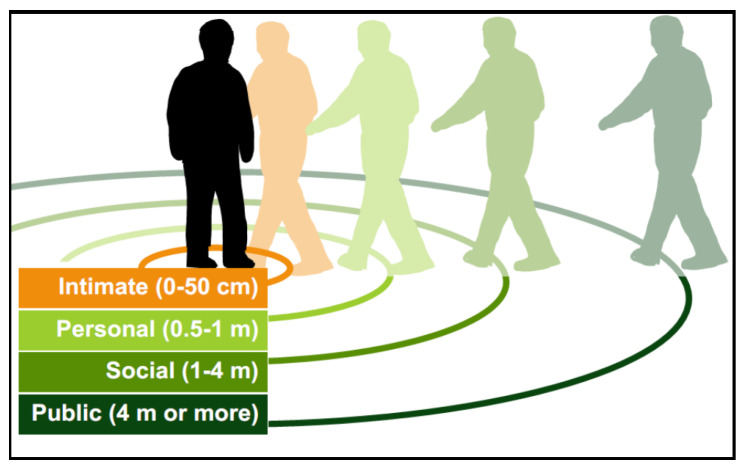
Interpersonal distances of people according Edward Hall’s proxemic theory (showing radius in meters) [24].

**Figure 2 micromachines-12-00193-f002:**
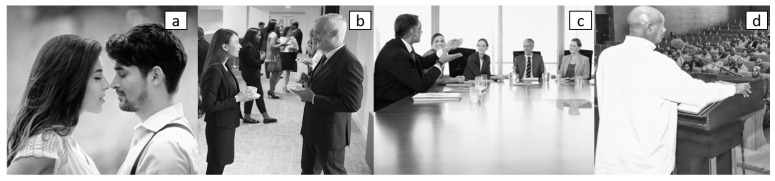
Interaction according to Proxemic zones: (**a**) intimate, (**b**) personal, (**c**) social and (**d**) public.

**Figure 3 micromachines-12-00193-f003:**
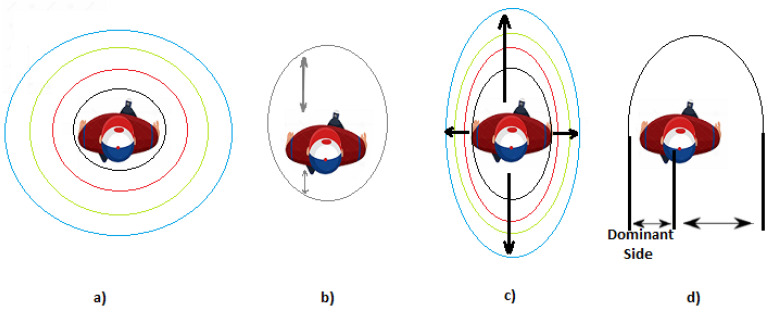
Different shapes of proxemic zones: (**a**) the classical four proxemic zones of Hall’s theory; (**b**) the personal space refers to the “private sphere” in the Social Force model; (**c**) proxemic zones as concentric ellipses; (**d**) proxemic zone is smaller on the dominant side of the pedestrian [22].

**Figure 4 micromachines-12-00193-f004:**
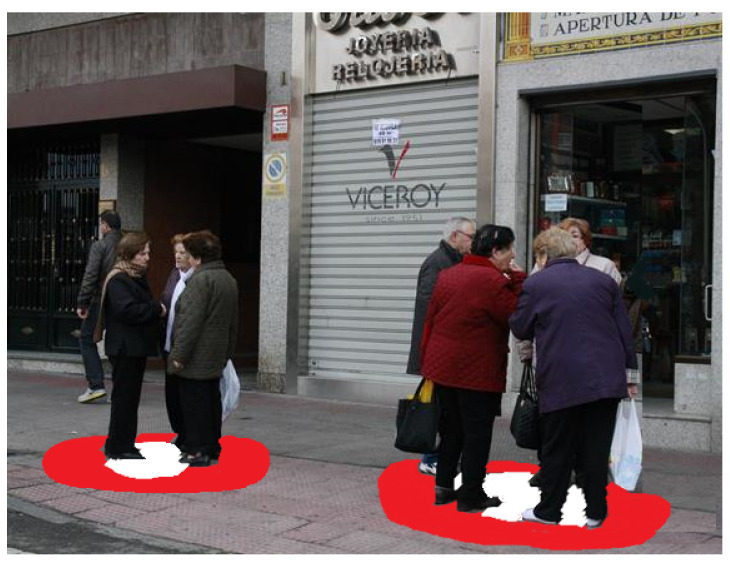
“O” and “P” spaces for groups (white and red, respectively).

**Figure 5 micromachines-12-00193-f005:**
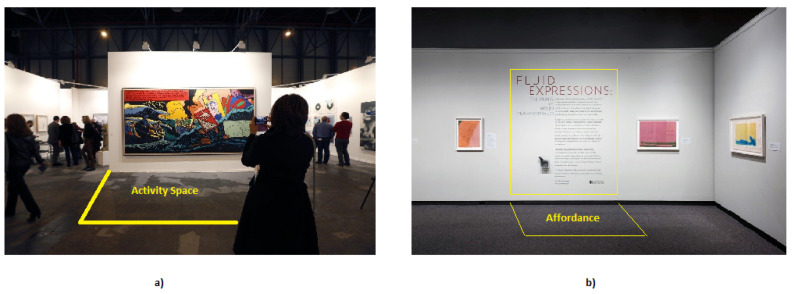
(**a**) Activity space and (**b**) affordance space.

**Figure 6 micromachines-12-00193-f006:**
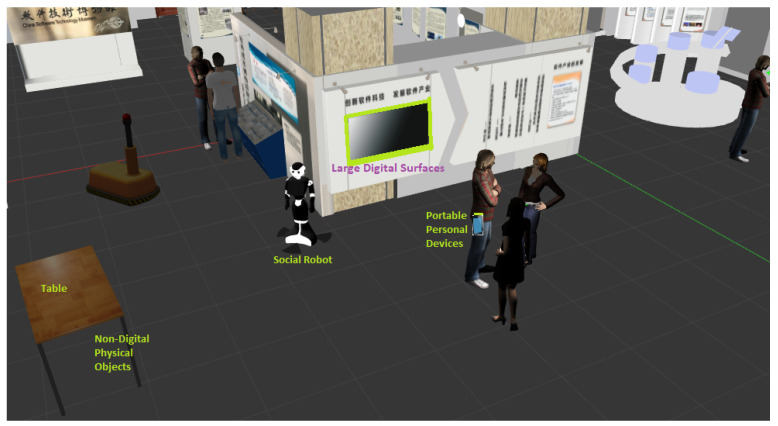
Proxemic interactions associate people to digital devices, digital devices to digital devices, and non-digital physical objects to both people and digital devices.

**Figure 7 micromachines-12-00193-f007:**
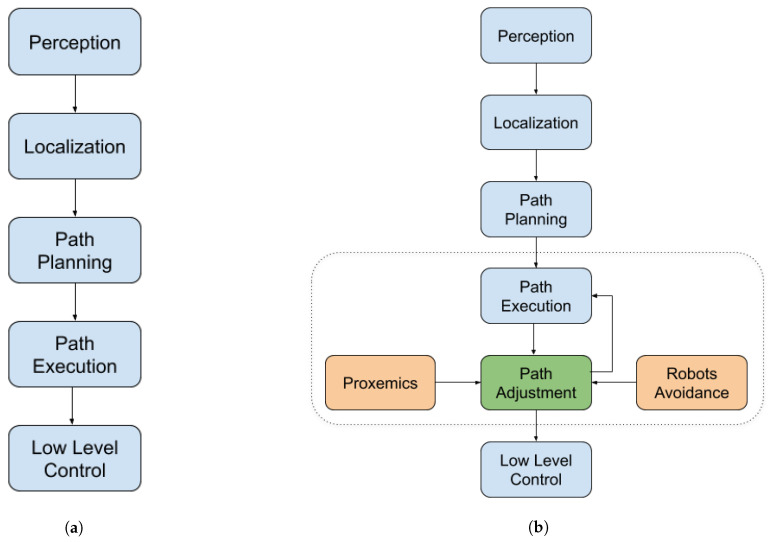
Navigation for autonomous robots comparison: (**a**) the traditional approach and (**b**) the proposed social navigation approach.

**Figure 8 micromachines-12-00193-f008:**
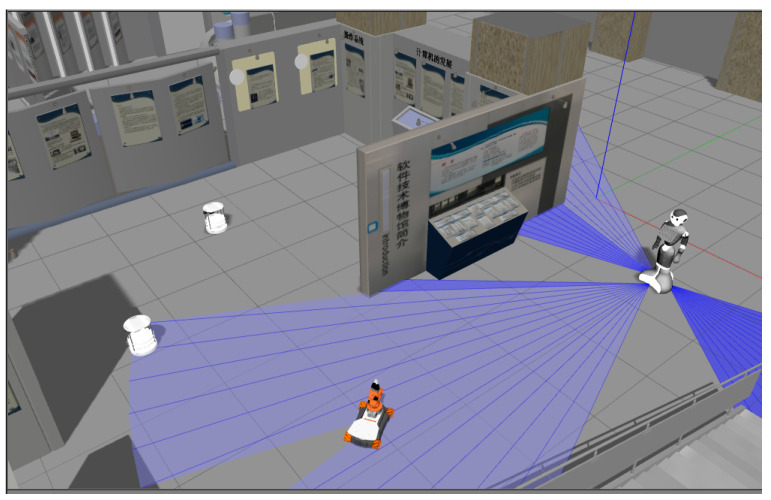
Group of robots navigating in the same environment.

**Figure 9 micromachines-12-00193-f009:**
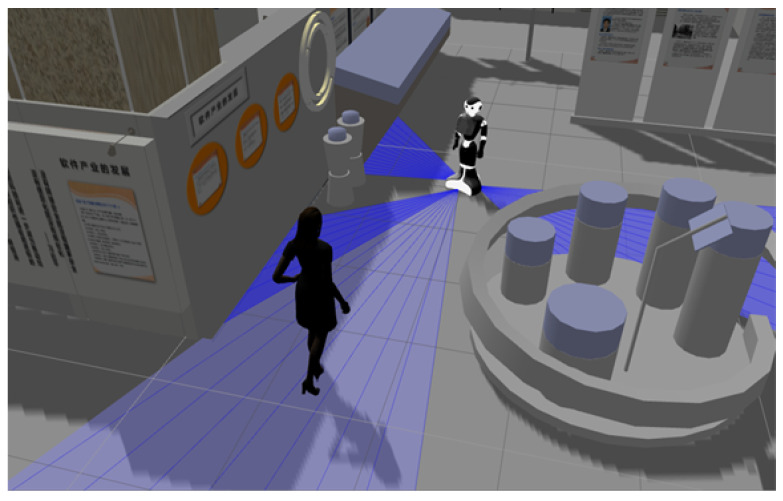
Robot in front to a human being.

**Figure 10 micromachines-12-00193-f010:**
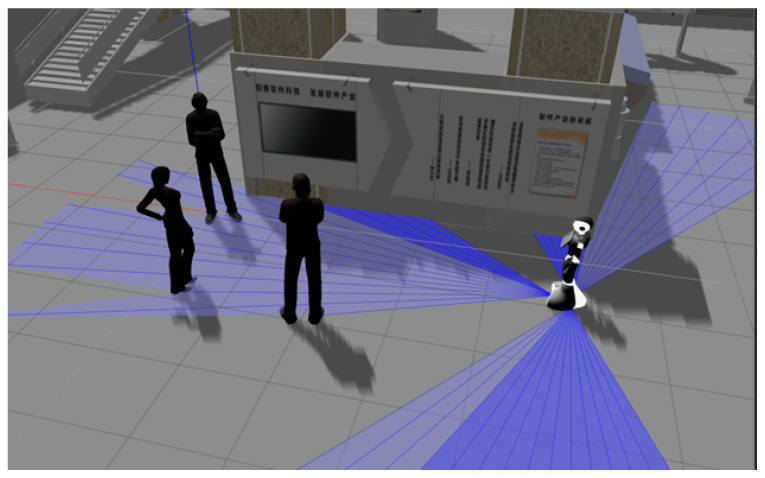
Robot in front to a group of humans.

**Figure 11 micromachines-12-00193-f011:**
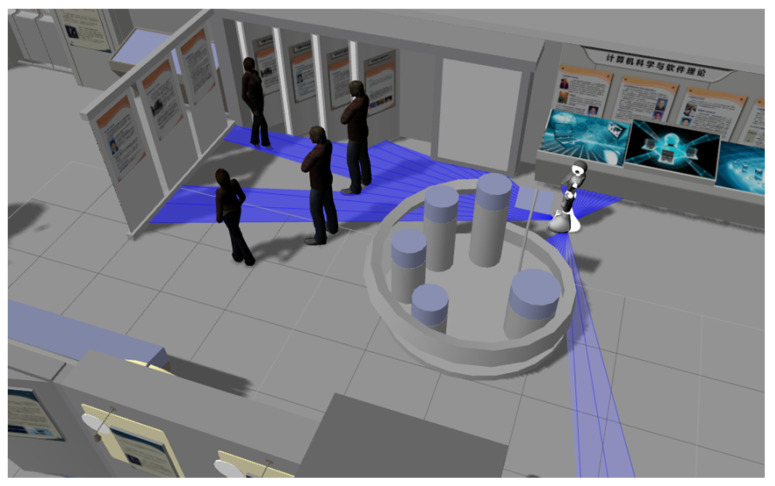
Group of people paying attention to one person.

**Figure 12 micromachines-12-00193-f012:**
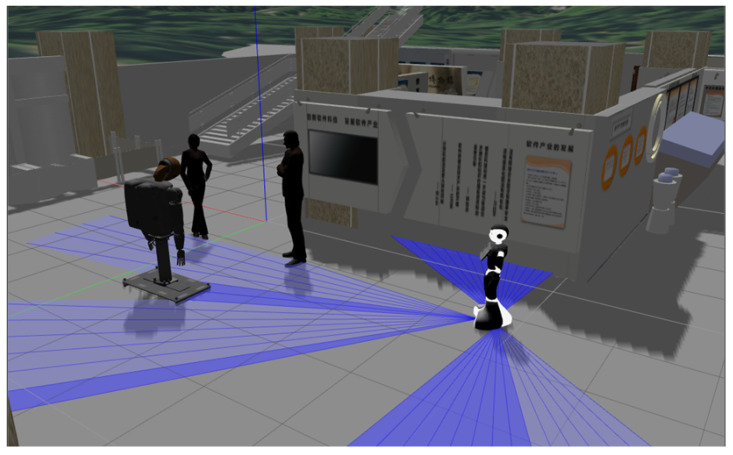
Group of persons with a robot.

**Figure 13 micromachines-12-00193-f013:**
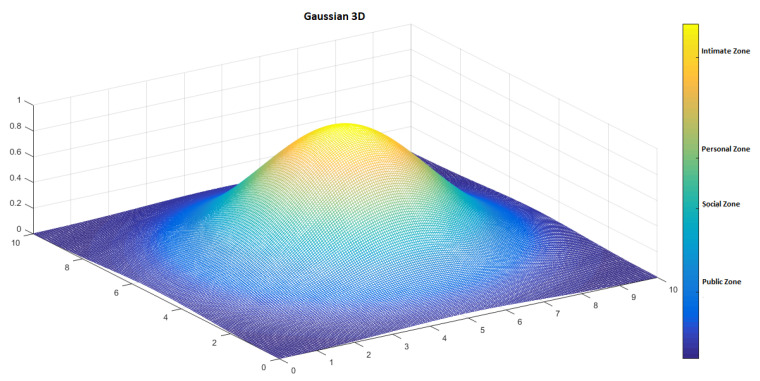
Gaussian in 3D.

**Figure 14 micromachines-12-00193-f014:**
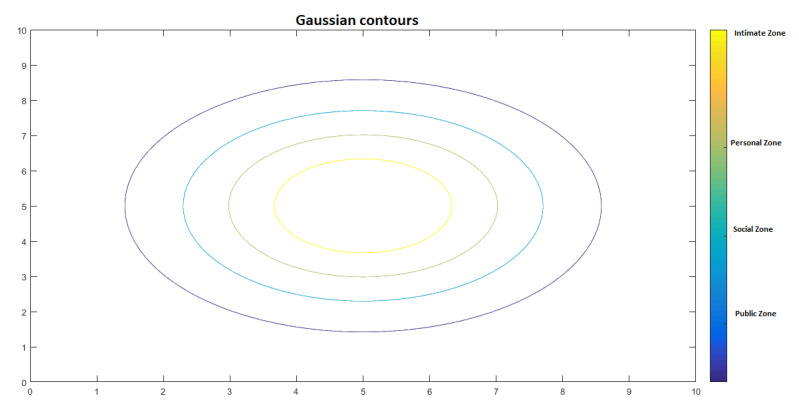
Representation of proxemic zones by a symmetrical Gaussian.

**Figure 15 micromachines-12-00193-f015:**
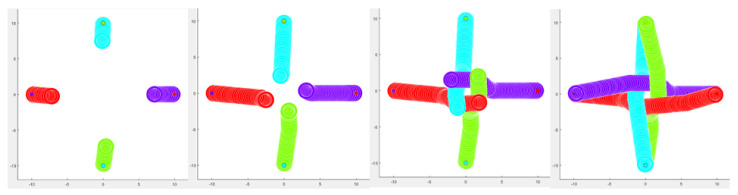
Path generation of the four robots, considering the evolution of the performance of the ORCA algorithm in a dynamic simulation environment without human presence.

**Figure 16 micromachines-12-00193-f016:**
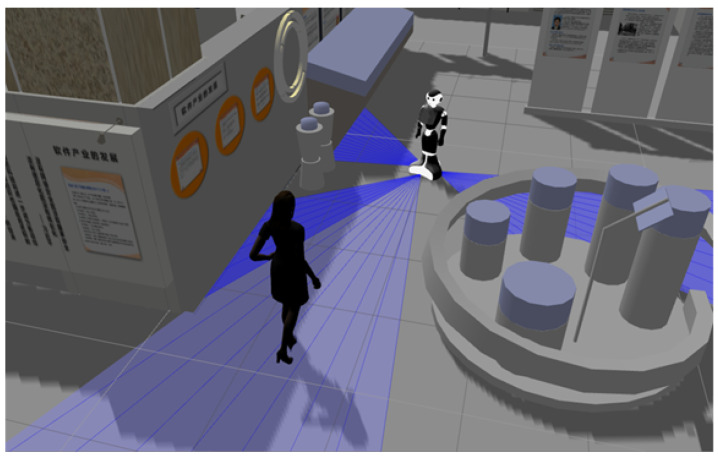
Environment with a human.

**Figure 17 micromachines-12-00193-f017:**
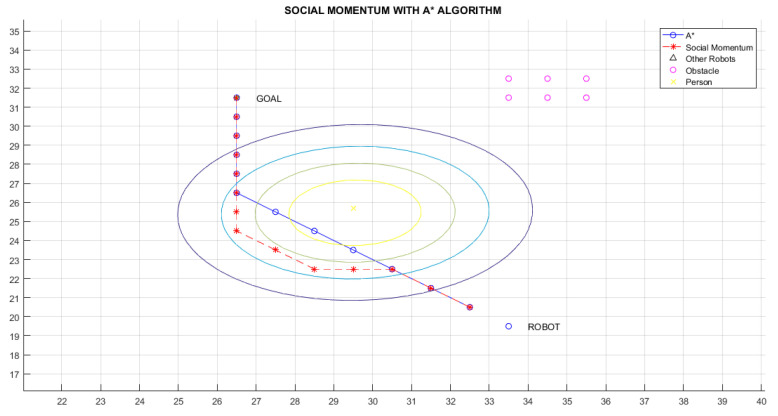
Path 1—Social Momentum Algorithm with A*.

**Figure 18 micromachines-12-00193-f018:**
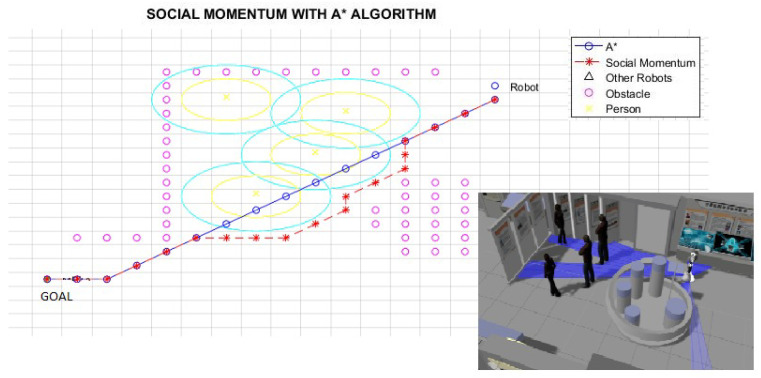
Path 2—Generation considering group of humans.

**Figure 19 micromachines-12-00193-f019:**
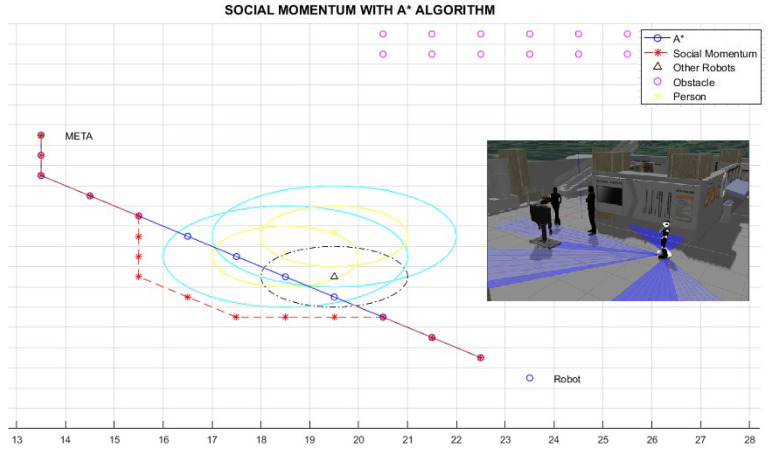
Path 3—Generated proxemic zone in a robot because is near to the proxemic zone of humans.

**Figure 20 micromachines-12-00193-f020:**
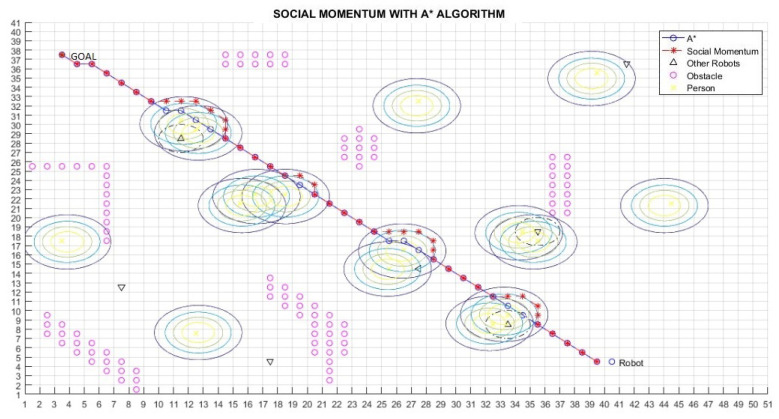
Path 4—Example 1 of navigation considering humans and robots in groups.

**Figure 21 micromachines-12-00193-f021:**
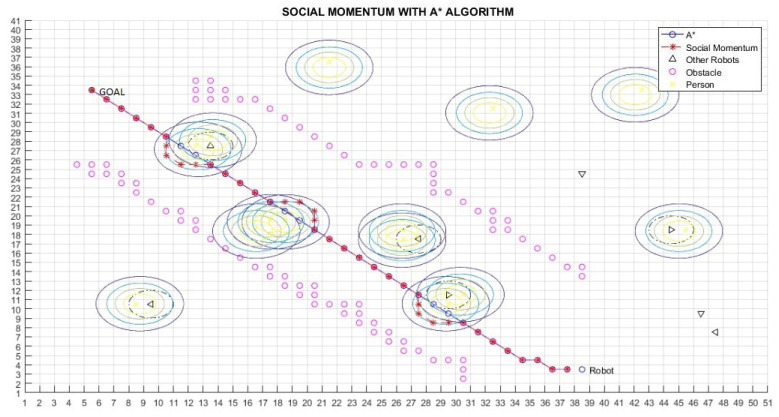
Path 5—Example 2 of navigation considering humans and robots in groups.

**Table 1 micromachines-12-00193-t001:** Comparative table of studies on social navigation based on proxemics.

Reference	Proxemic					
	Distance	Location	Movement	Orientation	Individual /Group	Proxemic Zone
J Rios et al. [22]	x	x	x	x	Both	4 zones
J. Mumm and B Mutlu [19]	x	x	x	-	Individual	Personal
C. Mavrogiannis et al. [52]	-	-	x	-	Group	-
C. Lobato et al. [55]	x	-	-	x	Individual	Intimate, Personal
E. Avrunin et al. [29]	x	x	x	x	Individual	Personal, Social
K. Zheng et al. [12]	-	x	x	-	Both	Personal
Maja Pantic et al. [51]	-	-	-	-	-	-
A. Vega-Magro et al. [61]	x	-	x	-	Group	4 zones
H. Khambhaita et al. [57]	-	x	-	x	-	-
Mead et al. [58]	x	-	-	x	-	-
D. Tokmurzina et al. [31]	x	-	-	-	Individual	-
J. Han and I. Bae [60]	x	x	-	x	-	-
P. Patompak et al. [62]	x	-	-	x	Individual	4 zones

**Table 2 micromachines-12-00193-t002:** $L_{ratio}$ and Smooth metrics.

Scenario	$L_{ratio}$	Smooth
Path 1 (Figure 17)	0.86	0.08 rad. ( $4.6^{\circ }$)
Path 2 (Figure 18)	0.84	0.38 rad. ( $21.8^{\circ }$)
Path 3 (Figure 19)	0.80	0.77 rad. (45${}^{\circ }$)
Path 4 (Figure 20)	0.84	0.42 rad. ($24.1^{\circ }$)
Path 5 (Figure 21)	0.85	0.51 rad. ($30^{\circ }$)
**Average**	0.84	0.43 rad (25${}^{\circ }$)

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
