# Peer review of "An Approach of Social Navigation Based on Proxemics for Crowded Environments of Humans and Robots"

_micromachines, 2021, doi:10.3390/mi12020193_

Round 1

Reviewer 1 Report

The paper brings interesting topic.

The main novelty is novel approach to adapt social robots navigation to different scenarios, by integrating proxemic interactions with traditional navigation algorithm.

The structure of the paper is very good.

Also introduction and analysis of problems are perfect. There is o comparation of Studies on Social Navigation based on Proxemics.

The proposal in this research is focused on the development of a navigation system for social robots acting in environments populated by both humans and robots. The proposed approach considers that the robot must avoid collisions in a social environment and incorporates restrictions based on proxemics, in particular for robot-human interaction for the greater comfort of people.

comments:

check the grammar!

  • Mistake: page 4 line 138: "pf people" - of people
  • page 6 line 190: "specif people" - specific
  • page 7 line 231: "robots.Among"

etc.

Depending on the focus, this paper is more suitable for Robotics (MDPI) or Sensors (MDPI). Micromachines magazine has a different scope. But that's just my opinion and it's decided by the topic editor.

I recommend to revise it.

Author Response

Response to Reviewer 1 Comments

Point 1: check the grammar!

Mistake: page 4 line 138: "pf people" - of people

page 6 line 190: "specif people" - specific

page 7 line 231: "robots.Among"

Response 1: The grammar has been changed as recommended.

Reviewer 2 Report

The introduction is clear and introduces the reader to the main research context, enhancing the navigation algorithm to be more natural for humans that interact with social robots. i.e. making social robots to behave/move as humans to avoid people feel unsafe. Despite the references are good, I miss more recent works as (except a preprint unreviewed work published in ArXiv in 2020) most of the works were published in 2018 or earlier. At this point, I would encourage authors to update the literature review and include more recent references. Also, the ArXiV preprints should be replaced with final peer-reviewed works (conferences and/or journals).

Despite the main contributions of this paper, authors state that they are proposing "a novel approach to adapt social robots navigation to different scenarios". Is the novel approach the only contribution of this work? Are other researchers proposing different approaches to the same goal? What is different in your approach with others already present in the literature. I encourage authors to provide a detailed list with the main contributions of this work and hihglight the differences with respect to other similar approaches already present in the literature. 

The related work includes almost 20 references and a table which provides a comparative study of the relevant related works. However, at this point more the authors have provided a long introduction without, for instance, providing a clear definition of "Social Momentum". The Social Momentum algorithms is widely used in the proposed method description which is the Section where my major concerns concentrate. 

The proposed method description is, in my opinion, vaguely described. There is not any formal algorithm which makes research reproducibility difficult to reach. Some instructions about the steps are provided by examples in three cases (no humans/robots, with humans and with humans and robots) but the ways to integrate the algorithms are not provided. At this point, I still miss a formal definition of the Social Momentum algorithm.

The experimental evaluation includes a simulation in each scenario. The experiments present visually how the robot behaves in one particular case, but metrics about the appropriateness of the proposed method with respect to others are not provided. 

In my opinion, the authors are somehow providing a proof-of-concept of their proposed social navigation algorithm. The idea of including the proxemic zones is interesting. However, the way it is presented may not be the most appropriate one. 

Minor:

Line 135-16: "Thus, it is also important to define proxemic zones for group pf people" Typo "pf people" --> "of people"

Regarding the use of external figures and photos, authors must ensure/check that they hold the rights to include them in the publication. 

Author Response

Response to Reviewer 2 Comments

Point 1: The introduction is clear and introduces the reader to the main research context, enhancing the navigation algorithm to be more natural for humans that interact with social robots. i.e. making social robots to behave/move as humans to avoid people feel unsafe. Despite the references are good, I miss more recent works as (except a preprint unreviewed work published in ArXiv in 2020) most of the works were published in 2018 or earlier. At this point, I would encourage authors to update the literature review and include more recent references. Also, the ArXiV preprints should be replaced with final peer-reviewed works (conferences and/or journals)

Response 1: The article has been updated and has 10 recent publications from 2019 to 2020.

Point 2: Despite the main contributions of this paper, authors state that they are proposing "a novel approach to adapt social robots navigation to different scenarios". Is the novel approach the only contribution of this work? Are other researchers proposing different approaches to the same goal? What is different in your approach with others already present in the literature. I encourage authors to provide a detailed list with the main contributions of this work and hihglight the differences with respect to other similar approaches already present in the literature.

Response 2: A better definition of contributions has been added in the Introduction and related works sections in Paper

Point 3:The related work includes almost 20 references and a table which provides a comparative study of the relevant related works. However, at this point more the authors have provided a long introduction without, for instance, providing a clear definition of "Social Momentum". The Social Momentum algorithms is widely used in the proposed method description which is the Section where my major concerns concentrate

Response 3: A better definition of the social momentum has been added to the related work section in Paper

Point 4: The proposed method description is, in my opinion, vaguely described. There is not any formal algorithm which makes research reproducibility difficult to reach. Some instructions about the steps are provided by examples in three cases (no humans/robots, with humans and with humans and robots) but the ways to integrate the algorithms are not provided. At this point, I still miss a formal definition of the Social Momentum algorithm.

Response 4: The proposed method description is more detailed by means of a Pseudocode in “Social navigation system: implementation and results" section.

Point 5: The experimental evaluation includes a simulation in each scenario. The experiments present visually how the robot behaves in one particular case, but metrics about the appropriateness of the proposed method with respect to others are not provided. 

Point 6: In my opinion, the authors are somehow providing a proof-of-concept of their proposed social navigation algorithm. The idea of including the proxemic zones is interesting. However, the way it is presented may not be the most appropriate one. 

Response 6: A better definition of contributions has been added in the Introduction and related works

sections in Paper

Point 7: Minor:

Line 135-16: "Thus, it is also important to define proxemic zones for group pf people" Typo "pf people" --> "of people"

Regarding the use of external figures and photos, authors must ensure/check that they hold the rights to include them in the publication. 

Response 7: All photos and figures are from original sources

Round 2

Reviewer 2 Report

The authors have done an effort to address all my comments except one. The experiments (simulations) visually present how the robot behaves in one particular case, but metrics about the appropriateness of the proposed method with respect to others are not provided. 

I would suggest authors to provide a metric (or set of metrics) to evaluate the proposed paths and, event, much longer ones. At every robot step, marked with the corresponding symbol on (for instance) Figure 30, you should update the corresponding counters. i.e., if the robot falls under the intimate zone of a person, the corresponding counter counting the cases falling under 'intimate' should be increased by one. How to deal when the robot falls under the zones of many people would still be an open issue to be addressed by the authors. You could consider all people around the robot or just the worst case.

What I would like to transmit is that a metric based on those counters could allow the method grading in one value (or in a vector) and ease the comparison, specially if the evaluated path is long. 

Author Response

Point 1: The authors have done an effort to address all my comments except one. The experiments (simulations) visually present how the robot behaves in one particular case, but metrics about the appropriateness of the proposed method with respect to others are not provided.
I would suggest authors to provide a metric (or set of metrics) to evaluate the proposed paths and, event, much longer ones. At every robot step, marked with the corresponding symbol on (for instance) Figure 30, you should update the corresponding counters. i.e., if the robot falls under the intimate zone of a person, the corresponding counter counting the cases falling under 'intimate' should be increased by one. How to deal when the robot falls under the zones of many people would still be an open issue to be addressed by the authors. You could consider all people around the robot or just the worst case.
What I would like to transmit is that a metric based on those counters could allow the method grading in one value (or in a vector) and ease the comparison, specially if the evaluated path is long.

Response 1: According to the reviewer's observations, we defined two metrics presented and developed in Section 5.6 of the paper. In all the implementations presented, the robot did not invade people's intimate zones, for this reason, the counter suggested by the reviewer was not implemented.

We also improved the explanation and presentation of Algorithm 1 to emphasize that Social Momentum assures to never invade intimate zones and for better understanding of the whole social navigation process

Round 3

Reviewer 2 Report

Many thanks for your answer, I appreciate the incorporation of metrics.

Probably, I did not express well my self in the previous round. The counter based metrics could be useful for comparisons. You can provide that a solution falls n times under intimate, m times under personal, k times under social and l times under public. As you said your proposal never fall insto personal whereas A* fall a few times. That vector (which could be shown as a Table) can ease the interpretation of the tables, specially when the experiments involves long paths and multiple methods to compare. However, the metric you have provided is also good.